# Emotional Recognition and Its Relation to Cognition, Mood and Fatigue in Relapsing–Remitting and Secondary Progressive Multiple Sclerosis

**DOI:** 10.3390/ijerph192416408

**Published:** 2022-12-07

**Authors:** Ornella Argento, Chiara Piacentini, Michela Bossa, Ugo Nocentini

**Affiliations:** 1Behavioral Neuropsychology Laboratory, I.R.C.C.S. “Santa Lucia” Foundation, 00179 Rome, Italy; 2Department of Clinical Sciences and Translational Medicine, University of Rome “Tor Vergata”, 00133 Rome, Italy

**Keywords:** emotional recognition, social cognition, cognition, mood, fatigue, multiple sclerosis

## Abstract

(1) Background: Emotional recognition (ER), the ability to read into others’ minds and recognize others’ emotional states, is important in social environment adaptation. Recently it has been found that ER difficulties affect patients with multiple sclerosis (pMS) and relate to different gray matter atrophy patterns from secondary progressive (SP-pMS) and relapsing–remitting (RR-pMS). The aim of this study was to compare the performances of the two MS phenotypes on the Reading the Mind in the Eyes test (RMEt) and other cognitive, mood and fatigue measures. We also examined associations between performance on the RMEt and cognitive, mood and fatigue variables. (2) Methods: A total of 43 pMS (27RR-pMS/16SP-pMS) underwent a clinical assessment, the RMEt, the cognitive battery, and completed mood and fatigue questionnaires. Both groups’ performances on the RMEt were then correlated with all these measures. (3) Results: the RMEt scores of RR-pMS were significantly correlated with the impairment degree in some cognitive scores. SP-pMS scores correlated mainly with fatigue, anxiety, anger and depression. (4) Conclusions: ER performances relate to cognitive aspects in RR-pMS, whereas mainly to mood outcomes in the SP-pMS group. We can hypothesize that deficits in ER are a further sign of disease progression. Our data support the different roles of cognitive and emotional deficits related to different disease courses and lesional correlates.

## 1. Introduction

Multiple sclerosis (MS) is a chronic inflammatory degenerative disease of the central nervous system. It is the main cause of non-traumatic disability among young adults and its course is highly variable and unpredictable. MS is characterized by focal and diffuse white matter damage, cortical lesions, cortical atrophy and microstructural abnormalities in deep gray matter (GM) that affect structural and functional connectivity between various brain regions [1,2].

From a clinical point of view, patients with MS (pMS) manifest a rather heterogeneous condition, characterized by cognitive, sensorimotor, visual, vegetative, cerebellar, psychological, social and emotional symptoms, which lead to functional disability and a reduced quality of life [3].

Among the mood disorders, anxiety and depression are commonly reported in pMS [4,5,6]. The first is associated with high levels of disability, progressive disease course [4], comorbid depressive symptoms [5,6] and female sex [5], while depression is associated with neuropathology [7,8], cognitive deficits [9,10] and poor social support [11].

Furthermore, from the emotional point of view, pMS may also have difficulty with anger management [12]. Benito-Leon and colleagues [13] observed higher levels of trait anger in pMS than in a healthy control group. However, other studies have not shown a clear increase in anger levels in MS compared to controls [12,14]; Nocentini et al. [12] reported that pMS had a greater tendency to withhold anger notwithstanding a lower level of control on it than the general population.

Fatigue is one of the most common symptoms experienced by up to 75–90% of pMS [15]. It is considered one of the most disabling symptoms of MS. It can drastically affect patients’ quality of life by also causing severe socioeconomic difficulties due to job loss [15].

Cognitive deficits affect 42–70% of people with MS with significant consequences on quality of life and social participation [16]. The most frequently compromised cognitive functions are memory, attention, information processing speed, abstract/conceptual reasoning and visuospatial skills; on the other hand, primary language skills, immediate and implicit memory and verbal intelligence seem less impaired [16,17]. Although the above-reported cognitive domains have been well studied in MS [18], little attention has been paid to social cognition (SC), defined as a multidimensional construct that encompasses the theory of mind (ToM), emotional recognition (ER) and empathy [19]. These processes allow humans to understand themselves and other individuals, to interact with them and to adaptively orient behaviors toward appropriate goals [20]. Interest in SC has grown over the past twenty years, to such an extent that in 2013 it was included in the latest revision of the Diagnostic and Statistical Manual of Mental Disorders as one of the six major neurocognitive domains, along with learning and memory, complex attention, executive function, perceptual–motor function and language.

In MS, studies conducted on SC have mainly focused on ER, which is the process of identifying human emotions from facial expressions [21], and on ToM, defined as the ability to decode and interpret the mental states of others and to use them to make inferences and predict their behaviors [22]. Two recent meta-analyses [23,24] confirmed the presence of significant deficits in ER and ToM in MS. In particular, in pMS, several researchers have observed difficulties in recognizing negative emotions [25,26,27] and in attributing mental states to others, in verbal and non-verbal ToM tasks [28,29,30]. SC deficits may be of similar magnitude to those observed in other cognitive domains [23] and may contribute to the interpersonal and psychosocial relationship difficulties complained of by pMS [31,32,33]. Aspects of SC (including ToM) and identification of others’ emotional expressions have also been reported to be disturbed in the pediatric onset of MS [34].

More recently, numerous studies have assessed the relationship between SC deficits and general cognitive impairment, revealing inconsistent results. For example, significant correlations have been reported between deficits in SC and processing speed [26,30,35,36,37], working memory [38,39,40,41] and problem-solving [37,41,42]. By contrast, others have found no relationship between social cognitive performance and general cognitive impairment [43,44,45,46]. Pitteri and colleagues [47] sought to overcome these contradictory results by studying performance in SC tasks in a group of relapsing–remitting MS patients (RR-pMS) without cognitive impairment. This study gave results in support of the independence of the SC from classic cognitive deficits: the RR-pMS, despite being cognitively preserved, showed a significantly lower performance than a paired group of healthy controls in ToM, ER and empathy tasks. The absence of association between SC and general cognitive functioning is also supported by a recent meta-analysis [48] conducted on 1708 pwMS and 1518 healthy controls.

Interestingly, in a previous study, we found that ER difficulties often affect MS patients with consequences on quality of life, psychosocial adaptation due to problems in understanding pragmatic language statements (such as irony), employment and interpersonal domains (such as personal relationships). ER difficulties were found to relate to a different GM atrophy pattern in secondary progressive (SP-pMS) and RR-pMS [49]. Furthermore, SP-pMS performed significantly worse than matched healthy controls in the emotional recognition task, while RR-pMS did not. We supposed that the different impairment in ER performance between RR-pMS and SP-pMS could depend on the demyelination and neurodegeneration of the cognitive component supporting the ability to process others’ mental states from facial expressions, in the initial stages of MS, and on the affective component in the secondary phase, following the work of Isernia and colleagues [50].

Considering these premises, we could speculate that a widespread pattern supporting SC functioning could have areas or circuits overlapping with those supporting other cognitive and mood functions. These overlapping areas or circuits, when interested in MS-related neurodegeneration, may deal with different patterns of impairment in each aspect of SC. If so, a clearer knowledge of the relationship between SC and other cognitive domains and mood aspects could be acquired, taking into consideration the above-reported different GM atrophy patterns in SP-pMS and RR-pMS. Therefore, the present study intends to explore how the already reported differences between the two main MS phenotypes relate to other important MS-related clinical aspects, such as cognition, mood and fatigue.

## 2. Materials and Methods

### 2.1. Participants

To consecutively be enrolled in this study, patients who attended the “Santa Lucia” Foundation MS Clinic had to satisfy the following inclusion criteria: (1) have a diagnosis of MS (McDonald’s criteria revised in 2011 by Polman [51]); (2) have an RR phenotype (MS in which patients have relapses of MS and periods of recovery and stability in between relapses) or SP phenotype (a progressive form of MS that involves fewer attacks, in which the disability gets steadily worse); (3) be 18–65 years old; (4) be native Italian speakers.

The exclusion criteria were: (1) a previous diagnosis of psychiatric or neurological disorders with the exception of MS or a severe systemic disease; (2) use of psychotropic drugs; (3) cognitive impairment severity interfering with the comprehension of tasks; (4) significant impairments in visual, auditory or linguistic functions interfering with the execution of tasks; (5) an MS relapse in the three months prior to their enrolment; (6) use of steroids over the previous month.

Due to the abovementioned criteria, the final sample of participants was 43 patients (27 RR-pMS and 16 SP-pMS). The principal demographic and clinical characteristics of all participants are summarized in Table 1. This study was approved by the Local Ethics Committee (CE/PROG.444-09), and all participants gave their written informed consent before taking part in the study.

### 2.2. Procedures

After having signed the informed consent, patients underwent a clinical assessment with the Expanded Disability Status Scale (EDSS; [52]) by a certified physician and a cognitive assessment with the Italian version of the Minimal Assessment of Cognitive Functioning in Multiple Sclerosis battery (MACFIMS; [53,54]). This battery is composed of seven tests: the California Verbal Learning Test-2 (measurement of verbal learning and memory; CVLT-2); the Brief Visuospatial Memory Test-Revised (visuospatial memory test; BVMTR); the Symbol Digit Modalities Test (information processing speed; SDMT; oral version); the Benton Judgment of Line Orientation test (measures the accuracy of spatial orientation judgments; BJLO); the Controlled Oral Word Association Test (a measure of phonemic fluency; COWAT); the Delis–Kaplan Executive Function System Sorting Test (D-KEFS; a measure of executive functions efficiency); and the Paced Auditory Serial Addition Test 3-s version (a measure of working memory; PASAT). A global cognitive impairment index was also computed for the variables included in the MACFIMS and the calculation was derived from Argento and colleagues [54].

The EDSS is a scale helpful for quantifying disability in MS and monitoring changes in the level of disability over time. The score is based on measures of impairment in eight functional systems: visual functions (problems with sight and visual field); brainstem functions (problems with speech, swallowing and nystagmus); pyramidal functions (muscle weakness or difficulty moving limbs); cerebellar functions (ataxia, loss of balance, coordination or tremor); sensory functions (numbness or loss of sensations); bowel and bladder functions; cerebral functions (problems with thinking and memory); ambulation. The EDSS scale ranges from 0 to 10, in 0.5-unit increments that represent higher levels of disability [52].

After that, pMS were asked to complete a mood assessment performed with the following questionnaires:

The State-Trait Anxiety Inventory (STAI-Y; [55]). This questionnaire is composed of two scales of 20 items each, referring to, respectively, state anxiety and trait anxiety. Total scores are calculated separately for two scales, with higher scores indicating higher anxiety severity.

The Beck Depression Inventory-fast screen (BDIfs; [56,57]). This is a seven-item questionnaire that assesses dysphoria, anhedonia, suicidal ideation and cognition-related symptoms on a three-point scale. The total score is calculated by summing the response given for each item, with higher scores indicating higher depression severity.

The Modified Impact Fatigue Scale (MFIS; [58]). This is a self-report questionnaire of 21 items rated on a 5-point scale (0 = “Never”; 4 = “Almost always”). The total MFIS score can range from 0 to 84 and is computed by adding the scores from the physical, cognitive and psychosocial subscales. Higher scores indicate greater fatigue.

The State-Trait Anger Expression Inventory-second edition (STAXI-II, [59]). This is a self-report questionnaire used for the assessment of anger levels and expressions. It is made up of 57 items rated on a four-point scale (1 = “Almost never”; 4 = “Almost always”). For each scale higher scores corresponded to higher levels of anger.

STAXI-II includes the seven scales described in Figure 1.

For the aims of the study, we collected data only on three subscales of the STAXI-II: state anger, trait anger and expression of anger index.

To conclude, patients performed a test specifically directed to the assessment of ER: the Reading the Mind in the Eyes test (RMEt; [60,61]). This test consists of 36 black and white stimuli representing the eye region of different individuals (men and women) and four possible adjectives for each stimulus. Participants must examine the images and choose, for each one, which of the four presented adjectives correctly describes the mental state that the person in the photo is expressing (See Figure 2). For each participant, the total number of stimuli correctly recognized was corrected for demographic variables, according to Serafin and Surian’s procedures [62].

### 2.3. Statistical Analysis

Demographic data were compared between RR-pMS and SP-pMS using independent sample *t*-tests for age and education, and a chi-squared test for gender (Table 1); *t*-tests were also used to test for group differences on cognitive and behavioral tests (Table 2). All cognitive and behavioral scores were corrected for the principal demographic variables (age, sex and education). No correction was made for clinical variables as the course diagnosis of MS already accounts for them. Both groups’ performances on the RMEt were then correlated with all these measures using Pearson correlation. For the purposes of this study, different types of statistical analysis were carried out using SPSS 18 (SPSS Inc., Chicago, IL, USA) and a significant threshold throughout this study will be set at *p* < 0.05.

## 3. Results

The results from the comparison of demographics between RR-pMS and SP-pMS revealed that the two samples were comparable in terms of sex and education, but differed significantly with regard to age, disease duration and EDSS score (Table 1), with SP-pMS having higher age, higher disability and higher disease duration.

Similarly, the comparison of cognitive, mood and fatigue scores (Table 2) proved significant differences between SP-pMS and RR-pMS for all cognitive scores except for BJLO (*p* > 0.05) with SP-pMS being more impaired than RR-pMS. A statistically significant difference was found between groups on fatigue (*p* = 0.001), though groups did not differ on anxiety, depression and anger (all *p*-values > 0.05). As expected, based on previous literature, the performances of RR-pMS and SP-pMS were significantly different (*p* = 0.005).

Analyses revealed that the RMEt scores of RR-pMS were significantly correlated with the impairment degree in some MACFIMS scores: CVLT-IR *(r* = −0.420; *p* = 0.029), D-KEFS card sorting 1 (*r* = −0.410; *p* = 0.034) and 2 (*r* = −0.488; *p* = 0.045), and the global degree of cognitive impairment (*r* = −0.431; *p* = 0.025; Table 3). The RMEt scores of SP-pMS correlated with the BVMT-IR score (*r* = −0.566; *p* = 0.018), but were also significantly correlated with fatigue (*r* = −0.692; *p* = 0.002), anxiety (STAI-Y1: *r* = −0.689, *p* = −0.002; STAI-Y2: *r* = −0.688, *p* = 0.002), depression (*r* = −0.687; *p* = 0.002), anger sa (*r* = −0.690; *p* = 0.002), anger ta (*r* = −0.683; *p* = 0.003) and anger Er/index (*r* = −0.691; *p* = 0.002; Table 3).

## 4. Discussion

Emotional recognition (ER), in terms of the ability to read into others’ minds and recognize others’ emotional states, plays an important role in social environment adaptation.

Recently, the efficiency of this SC component has gained wide importance as the recent COVID-19 pandemic has imposed the use of face masks. Within even basic conversations, this has required the ability to understand others’ mental states only by observing the eye regions, making the maintenance of appropriate social relationships for pMS even harder. In fact, as previously demonstrated, these patients have difficulties in ER explored by the RMEt to a different degree for the two main courses of the pathology [49].

The aim of this study was to understand how the already reported differences between the two main MS phenotypes relate to other important MS-related clinical aspects, such as cognition, mood and fatigue.

The differences found in the comparison of the two samples for clinical and demographic variables are those expected as the SP course follows the RR one. This implies a longer disease duration and a higher disability level in SP-pMS compared to RR-pMS. In addition, the differences found for cognitive scores and for cognitive impairment index reflect the higher disability of SP-pMS as it has been widely reported in the literature [53,63]. Furthermore, as found in a previous study, patients in the two courses also had significantly different RMEt scores, with RR-pMS performing better than SP-pMS [49].

The results of this study offer a possible explanation of the previously found pattern of differences in emotional recognition between RR-pMS and SP-pMS. In this study, ER performances relate mainly to cognitive aspects in RR-pMS: in particular, immediate recollection of verbal memory and abstract reasoning, two components that require the efficiency of executive functions in terms of attention and categorization [64,65]. The 16 words that compose the CVLT-2 pertain to four categories: this influences the performance according to how subjects can be facilitated in the memorization by the categorization process; this is the same process used in the execution of the D-KEFS card sorting tasks [66]. Furthermore, some studies on alexithymia (defined as the poor efficiency in inferring others’ emotions) have found an association with executive functions’ efficiency (EF) and between EF and memory [67].

Conversely, in SP-MS, ER performances are related mainly to mood aspects, especially with perceived fatigue, anxiety and anger management aspects. The relationship found between fatigue and RMEt scores is not new in the literature. A significant relationship was already found in a previous study on SC in pMS [43], in which the authors hypothesized a similar anatomopathological pattern for the two aspects. Following their explanation, fatigue and SC impairment are associated with reduced function of the right prefrontal and anterior cingulate cortex and structural loss of fiber integrity in the frontal white matter [43]. In our study, we found that this overlapping is specifically related to the SP course of MS.

Interestingly, a significant correlation was found between the performance on the RMEt and the levels of state and trait anger in SP-pMS: patients with higher levels of state anger, trait anger and global anger expression (Er-Index) have a lower ability to decode others’ emotions. Following the description by Spielberger [59], people with high scores in the Er-Index tend to experience strong feelings of anger, which can be either repressed or expressed through aggressive behaviors. These people display anger in many shades of behavior, probably experience extreme difficulty in establishing interpersonal relationships and are at risk of developing medical disorders [59]. This seems to be the case for MS patients and in particular, following actual results, for SP-pMS. Therefore, we can hypothesize that patients in the SP course have some emotional dysregulation, making them impaired in both managing their intense emotions and recognizing others.

The data of this study allow us to say that at least for ER (as a component of the SC) it seems that the impairment found in the two main courses of MS could be caused by the damage of different circuits. In general, given the correlations with some cognitive scales, we can argue that the efficiency of some components of the so-called “executive functions” seems to be necessary to correctly perform the RMEt. In addition to the integrity of the visual system, a categorization process could be needed in order to facilitate the comparison with previous knowledge stored in the memory and subsequently, to allow the correct labeling of that emotion.

This study has some limitations that should be considered. First of all, we have not included a control sample to support the results found within our sample. However, we felt that since we had already found a difference between healthy subjects and MS patients in a previous study, we did not need an additional control group. We also did not perform a regression analysis with all RMEt-associated variables included to find a regression model and did not choose a more conservative approach in interpreting the results. However, we believed that our sample size did not allow for this type of analysis and would have led to underestimating an effect that is present but would only be seen with a less conservative approach. Future research will try to overcome these limitations.

In this study, we primarily aimed to explore the possibility that different MS courses may show different impairments in different aspects of the emotional decoding process. We started with ER but expected to explore these eventual differences in other components of the SC.

We strongly expect that this can help clinicians and researchers resolve the debate on considering SC as a specific cognitive domain or secondary to the efficiency of other cognitive processes.

On the other side, this can help clinicians to better understand when patients could benefit the most from specific training on SC. Otherwise, they should be addressed to other clinical approaches (such as psychotherapy, mindfulness-based programs, etc.) when their social concerns are more linked to mood aspects (such as anxiety, depression and/or anger management dyscontrol).

Future studies are needed to explore the possibility that even other components of SC can follow the pattern of ER and to understand if we can consider this deficit in ER as a further sign of disease progression in MS.

## 5. Conclusions

To conclude, the results of this study, in line with what has been demonstrated before, support the hypothesis of a different role of deficits in cognitive and emotional aspects of the ER process depending on MS course and differences in lesion burden. This means that SC, and in particular the emotional recognition process, may be correlated with the damage of different circuits associated with RMEt performance emerging in the two courses of the disease. Therefore, we could state that patients in the two MS courses are differently impaired even in SC performances.

## Figures and Tables

**Figure 1 ijerph-19-16408-f001:**
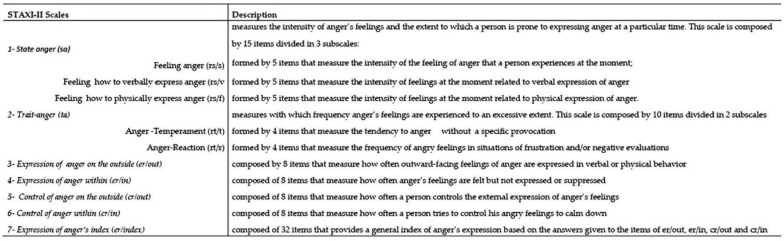
STAXI-II questionnaire. Description of each scale.

**Figure 2 ijerph-19-16408-f002:**
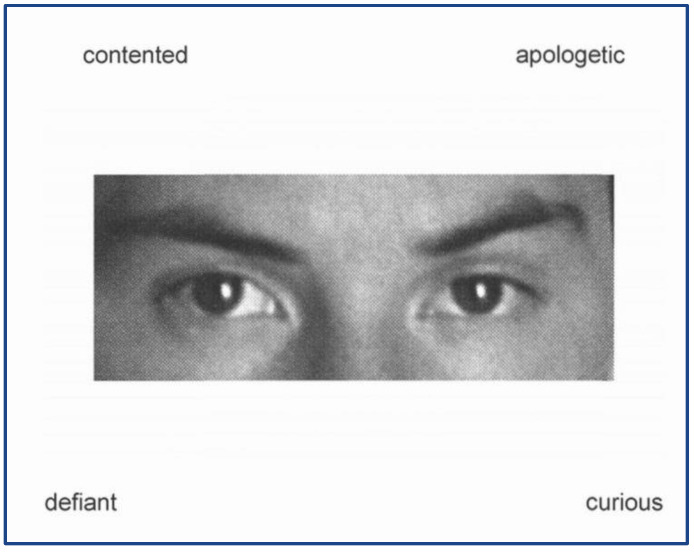
Example figure of Reading the Mind test item.

**Table 1 ijerph-19-16408-t001:** Clinical and demographic data of the MS sample.

	**RR-pMS**	**SP-pMS**	**Sig.**
Sex (m/f)	10/17	10/7	0.166
Age (M ± SD)	39.1 ± 9.1	50.9 ± 8.5	0.000
Education (M ± SD)	14.7 ± 2.4	13.9 ± 4.1	0.578
DD (M ± SD)	7.4 ± 6.8	17.3 ± 10.5	0.002
EDSS (M ± SD)	2.2 ± 0.8	5 ± 1	0.000

Abbreviations: m, male; f, female; M, mean; SD, standard deviation; EDSS, Expanded Disability Status Scale; DD, disease duration; RR-pMS, relapsing–remitting patients with multiple sclerosis; SP-pMS, secondary progressive patients with multiple sclerosis.

**Table 2 ijerph-19-16408-t002:** Cognitive and mood assessment, and comparison between RR-pMS and SP-pMS based on *t*-test analysis.

	RR-pMS (M ± SD)	SP-pMS (M ± SD)	Sig.		RR-pMS (M ± SD)	SP-pMS (M ± SD)	Sig.
CVLT-2_IR	54.3 ± 8.7	38.3 ± 10.9	0.000	BDIfs	1.9 ± 1.9	2.7 ± 2.9	>0.05
CVLT-2_DR	10.9 ± 3.3	7.5 ± 3.1	0.003	STAI-Y1	38.1 ± 9.6	42.8 ± 11.1	>0.05
BVMTR_IR	25.6 ± 6.0	16.4 ± 7.3	0.000	STAI-Y2	36.9 ± 9.7	42.6 ± 9.4	>0.05
BVMTR_DR	9.4 ± 2.1	6.7 ± 3.0	0.005	MFIS	23.8 ± 15.5	42.7 ± 15.6	0.001
SDMT	58.0 ± 10.1	31.2 ± 11.7	0.000	STAXI-sa	15.9 ± 1.8	19.7 ± 7.9	>0.05
DKEFS_C	10.4 ± 2.9	6.5 ± 2.6	0.000	STAXI-ta	16.1 ± 5.0	17.8 ± 3.6	>0.05
DKEFS_S	33.2 ± 17.3	19.6 ± 12.5	0.006	STAXI-Er/index	31.6 ± 13.5	33.3 ± 9.5	>0.05
BJLO	26.4 ± 2.7	23.0 ± 5.9	>0.05	RMEt	23.04 ± 4.1	18.92 ± 5.01	0.005
COWAT	41.0 ± 9.4	28.8 ± 9.5	0.000				
PASAT	47.9 ± 8.9	30.0 ± 11.6	0.000				
CII	6.4 ± 5.2	18.3 ± 6.2	0.000				

Abbreviations: pMS, patients with multiple sclerosis; M, mean; SD, standard deviation; CVLT-2_IR, California Verbal Learning Test-second edition_immediate recall; CVLT-2_DR, California Verbal Learning Test-second edition_delayed recall; BVMTR_IR, Brief Visuospatial Memory Test-Revised_immediate recall; BVMTR_DR, Brief Visuospatial Memory Test-Revised_delayed recall; SDMT, Symbol Digit Modalities Test; DKEFS_C, Delis–Kaplan Executive Functions System_total card sorting; DKEFS_S, Delis–Kaplan Executive Functions System_total score sorting; BJLO, Judgement of Line Orientation test; COWAT, Controlled Oral Word Association Test; PASAT, Paced Auditory Serial Addition Test; CII, cognitive impairment index; BDIfs, Beck Depression Inventory-fast screen; Stai-Y1/Y2, State-Trait Anxiety Inventory form Y1 and Y2; MFIS, Modified Impact Fatigue Scale; STAXI, State-Trait Anger Expression Inventory, (sa, state anger; ta, trait anger; Er/index, expression anger index); RMEt, Reading the Mind in the Eyes test.

**Table 3 ijerph-19-16408-t003:** Pearson correlation coefficient and effect size between RMEt performance and cognitive and mood aspects for both RR-pMS and SP-pMS.

	RR-pMS	SP-pMS		RR-pMS	SP-pMS
CVLT-2_IR	**−0.420** **(0.029)**	>0.05	BDIfs	>0.05	**−0.687** **(0.002)**
CVLT-2_DR	>0.05	>0.05	STAI-Y1	>0.05	**−0.689** **(0.002)**
BVMTR_IR	>0.05	**−0.566** **(0.018)**	STAI-Y2	>0.05	**−0.688** **(0.002)**
BVMTR_DR	>0.05	>0.05	MFIS	>0.05	**−0.692** **(0.002)**
SDMT	>0.05	>0.05	STAXI-sa	>0.05	**−0.690** **(0.002)**
DKEFS_cs1	**−0.410** **(0.034)**	>0.05	STAXI-ta	>0.05	**−0.683** **(0.003)**
DKEFS_cs2	**−0.488** **(0.045)**	>0.05	STAXI-Er/index	>0.05	**−0.691** **(0.002)**
BJLO	>0.05	>0.05			
COWAT	>0.05	>0.05			
PASAT	>0.05	>0.05			
CII	**−0.431** **(0.025)**	>0.05			

Abbreviations: pMS, patients with multiple sclerosis; M, mean; SD, standard deviation; CVLT-2_IR, California Verbal Learning Test-second edition_immediate recall; CVLT-2_DR, California Verbal Learning Test-second edition_delayed recall; BVMTR_IR, Brief Visuospatial Memory Test-Revised_immediate recall; BVMTR_DR, Brief Visuospatial Memory Test-Revised_delayed recall; SDMT, Symbol Digit Modalities Test; DKEFS_C, Delis–Kaplan Executive Functions System_total card sorting; DKEFS_S, Delis–Kaplan Executive Functions System_total score; BJLO, Judgement of Line Orientation test; COWAT, Controlled Oral Word Association Test; PASAT, Paced Auditory Serial Addition Test; CII, cognitive impairment index; BDIfs, Beck Depression Inventory-fast screen; Stai-Y1/Y2, State-Trait Anxiety Inventory form Y1 and Y2; MFIS, Modified Impact Fatigue Scale; STAXI, State-Trait Anger Expression Inventory, (sa, state anger; ta, trait anger; Er/index, expression anger index).

## Data Availability

Not applicable.

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
