# Peer review of "Emotional Recognition and Its Relation to Cognition, Mood and Fatigue in Relapsing–Remitting and Secondary Progressive Multiple Sclerosis"

_ijerph, 2022, doi:10.3390/ijerph192416408_

Round 1

Reviewer 1 Report

Manuscript entitled: "Emotional recognition and its relation to cognition, mood and fatigue in Relapsing-Remitting and Secondary-Progressive Multiple Sclerosis" is an interesting research article. It could be published after revision. There are some mistakes and inconsistencies:

- lack of control group

- all abbreviations should be explained after the first use, also in the Abstract

- explanation / definition of relapsing-remitting and secondary progressive MS types should be given

- EDSS scale should be described.

There are also some spelling and grammar mistakes that require correction.

Author Response

Reviewer 1:

Manuscript entitled: "Emotional recognition and its relation to cognition, mood and fatigue in Relapsing-Remitting and Secondary-Progressive Multiple Sclerosis" is an interesting research article. It could be published after revision.

We would like to thank the reviewer for his/her appreciation of our work and for his/her careful revision.

 There are some mistakes and inconsistencies:

- lack of control group

This issue has been considered within the limitations of the study. Thank you for suggesting it.

- all abbreviations should be explained after the first use, also in the Abstract

We have carefully revised the acronym across the manuscript and removed the ones that were non necessary, following reviewers suggestion.

- explanation / definition of relapsing-remitting and secondary progressive MS types should be given

We have made a description of the phenotypes in the inclusion criteria.

- EDSS scale should be described.

Following the reviewer's suggestion we provided a description of this scale in the section focused on the “procedures”.

There are also some spelling and grammar mistakes that require correction.

We have carefully revised the manuscript in order to correct the mistakes.

Reviewer 2 Report

Aims and hypotheses should be better described, both from linguistic and conceptual points of view.

Inclusion and exclusion criteria should be reported in clearer way and the sample characteristics should be reported after it.

The data reported are not very clear. Generally, throughout the paper the authors used a lot of abbreviation, which makes it difficult to read it and appreciate all the information reported. The same is true in the result section. Furthermore, table 2 is ambiguous since it’s not clear if the values reported are always p values (as it seems) or R values (actually, in the paper it was not stated what kind of correlation was performed and it should be specified). If table 2 reports only p values, then the R values are needed in order to understand the effect size of the correlations (that would be good to report too). Indeed, even when correlations are significant it is important to know their effect size before discussing data and drawing conclusions.

It would be nice also to report if the two groups of participants (RR and SP) reported significant differences in Mind in the Eyes test scores.

I feel that further comments and considerations, in particular about discussion, cannot be included in my report without clarifications related to above points. It is difficult to understand if the authors’ conclusions are properly derived by the data reported.

Aims and hypotheses should be better described, both from linguistic and conceptual points of view.

Inclusion and exclusion criteria should be reported in clearer way and the sample characteristics should be reported after it.

The data reported are not very clear. Generally, throughout the paper the authors used a lot of abbreviation, which makes it difficult to read it and appreciate all the information reported. The same is true in the result section. Furthermore, table 2 is ambiguous since it’s not clear if the values reported are always p values (as it seems) or R values (actually, in the paper it was not stated what kind of correlation was performed and it should be specified). If table 2 reports only p values, then the R values are needed in order to understand the effect size of the correlations (that would be good to report too). Indeed, even when correlations are significant it is important to know their effect size before discussing data and drawing conclusions.

It would be nice also to report if the two groups of participants (RR and SP) reported significant differences in Mind in the Eyes test scores.

I feel that further comments and considerations, in particular about discussion, cannot be included in my report without clarifications related to above points. It is difficult to understand if the authors’ conclusions are properly derived by the data reported.

Author Response

Reviewer 2:

We would like to thank the reviewer for his/her appreciation of our work and for his/her careful revision.

Aims and hypotheses should be better described, both from linguistic and conceptual points of view.

We tried to better describe aims and hypotheses of the actual study.

Inclusion and exclusion criteria should be reported in a clearer way and the sample characteristics should be reported after it.

We have modified the inclusion and exclusion criteria and reported the sample characteristics, as suggested.

The data reported are not very clear. Generally, throughout the paper the authors used a lot of abbreviations, which makes it difficult to read it and appreciate all the information reported. The same is true in the result section. Furthermore, table 2 is ambiguous since it’s not clear if the values reported are always p values (as it seems) or R values (actually, in the paper it was not stated what kind of correlation was performed and it should be specified). If table 2 reports only p values, then the R values are needed in order to understand the effect size of the correlations (that would be good to report too). Indeed, even when correlations are significant it is important to know their effect size before discussing data and drawing conclusions.

We agree with the reviewer about the number of abbreviations. We have tried to remove the ones that were not necessary.

We have also modified the Table about the Pearson’s correlation, adding the R values to provide information on the effect size.

It would be nice also to report if the two groups of participants (RR and SP) reported significant differences in Mind in the Eyes test scores.

We would like to thank the reviewer for his/her suggestion. We have included this information in table 2 and discussed it in the discussion.

I feel that further comments and considerations, in particular about discussion, cannot be included in my report without clarifications related to above points. It is difficult to understand if the authors’ conclusions are properly derived by the data reported.

Author Response

Reviewer 3:

Comments to the Author

This study sought to compare two MS phenotypes (i.e., RRMS and SPMS) on cognitive and mood outcomes including ER (the latter as measured by the Reading the Mind in the Eyes test). The authors reported independent associations by group between performance on the RMEt and all cognitive and mood  outcomes. Groups were well matched for sex and education but differed on sex, disease duration, and EDSS. The SPMS group had poorer performance on all cognitive measures relative to the RRMS group, except for BJLO. Groups differed significantly on fatigue with the SPMS group reporting greater fatigue. No significant group differences were found for anxiety, depression and anger.

We would like to thank the reviewer for his/her appreciation of our work and for his/her careful revision.

Main feedback:

General:

citations specific to each measure should immediately follow each mention of the acronym

We have carefully revised the acronym across the manuscript and removed the ones that were non necessary following the reviewer suggestion.

discussion section can be renamed “Conclusions” in alignment with the journals formatting

We have followed the IJERPH template we were provided with and included a Discussion section (point 4 ) and then the Conclusion section (point 5) as reported in the template.

Abstract:

The aim of the study is unclear as written. I would be more explicit to state that the goal of the present research is to compare two MS phenotypes on the Reading the Mind in the Eyes test and other cognitive measures and assess for associations with measures of mood and fatigue.

Following reviewer suggestions we tried to resentence the aim of the study in the abstract.

The term “emotional dysfunction” is confusing and has been used more in the context of describing externalizing behaviour. I feel similarly about “emotional deficits” and would keep to one term (i.e., ER) throughout the manuscript.

According to reviewers’ suggestions we modified these concepts across the manuscript.

Line 22: I would say “and” instead of “but” (i.e., “SPMSp scores correlated with the BVMTRir score, and were also significantly…”)

We followed reviewers’ suggestions.

Conclusions: I would be more explicit here (e.g., “Emotional recognition performance as measured by the RMEt was found to relate only to cognitive aspects in our RRMSp group, whereas it associated with mainly mood outcomes in our SPMSp group.”)

We modified the sentence according to the reviewer 's preference.

Introduction:

Adding more commentary/review of the literature on associations between anxiety, depression, and anger with ER in MS and/or in other populations will provide a helpful context for your work.

We agree with the reviewer about the necessity of including other literature in this field. However we are forced not to include other references as we are limited in the number of references allowed by guidelines.

Line 39: consider substituting “psycho-emotional” with “socio-emotional” or a similar term more commonly used in the literature

Following reviewers’ suggestion

Line 41: consider improving phrasing here by rewording to “Anxiety and depression are commonly reported in pMS.” A citation is also needed here.

We rephrased the sentence following reviewers’ suggestion and added the references.

Line 52: instead of “of them” consider rephrasing this sentence to reflect that 75-90% of pMS experience fatigue and then add a citation immediately following this sentence

We rephrased the sentence following reviewers’ suggestion and added the references.

Line 62: remove “the” before theory of mind

We removed the before theory of mind.

Line 72: there is an unnumbered citation here and a spelling error in meta-analyses

We removed the citation and corrected the error in meta-analyses.

SC and ER are not plural (e.g., Line 79 & Line 91)

We modified it as suggested by the reviewer.

Line 92: elaboration on how ER difficulties affect MS patients in your work would be helpful

According to the reviewer 's suggestion we included a brief overview of aspects impaired in MS patients as a consequence of ER difficulties.

Line 95: the use of the word “involvement” could be eliminated as you’re describing differences in performance – the explanation you have drawn regarding differences in ER performance in your earlier work, flow nicely into the aims of the present research though I am unclear what you mean by “the deterioration of the cognitive component supporting…”. Are you referring to a biological correlate (i.e., neurodegeneration in particular brain regions/networks)? See Line 96

Following the reviewer’ suggestion we tried to rephrase the sentence in order to make our intention clearer.

A main concern is the causal conclusion that “at least for ER (as a component of the SC) it seems that the impairment found on the two main courses of MS is caused by the damage of different circuits.” While brain-behaviour associations are described in your previously published 2022 manuscript, I would recommend that the conclusion made here be reframed as circuitry damage being associated with REMt performance.

As suggested by the reviewer we have rephrased the sentence about this aspect.

An interesting analysis would be to include all variables associated with REMt in a regression model to identify the relative strengths of the various predictors holding all others constant.

We agree with the reviewer, however we believe that the sample size did not allow us to perform this analysis. We would like to consider the possibility to increase the sample size and then explore this aspect.

Overall, you have summarized the general MS literature by domain nicely. However, I think the discussion around the specific ER literature in MS needs more context (i.e., noting the type of ER measures (basic ER vs. higher-order ToM) that have yielded differences between patients and controls in prior studies and have/have not shown associations with other cognitive measures. As well, more explanation of the role of ER on “environmental adaptation” is needed in order to clarify points raised later in the paper.

We agree with the reviewer on the usefulness of including more literature on many aspects of interest related to ER, ToM and cognition in general. However we are limited by the excessive length of the manuscript in its present form and the limited number of references allowed. We will insert this as a limit of the study and possible further study. However, further explanations on the role of the ER in environmental adaptation have been included in the introduction.

Further, on page 2, in your review of the literature, it may be helpful to bring in a brief overview of results in this area in pediatric-onset MS studies as they are mainly RRMS studies (i.e., studies that have looked at social and communication skills, ToM  and ER and their associations with other aspects of cognition and MRI variables such as Charvet et al., 2014; Green et al., 2018; Fabri et al., 2021)

We have included a brief statement on the presence of these difficulties in pediatric onset MS as well, however we could not include other references as we are limited in number and far beyond the allowable number.

Line 113: substitute “under” with “taking”

We have modified the inclusion and exclusion criteria, as suggested by another reviewer. The sentence with the word “under” was modified.

Methods:

Consider presenting the STAXI-II scales in a table

We would like to thank the author for his/her interesting suggestion. We have presented the STAXI-II in a figure/table and removed the description within the manuscript.

Line 189: add the specific adjectives

We have added this information.

When was EDSS assessed?

We provided to insert more information on EDSS following the reviewer's suggestion.

Statistical analyses:

 Line 202-205: “…were then correlated…” specify if your primary aim statistical analyses were done using Pearson or Spearman correlation, regression analyses and if there were any covariates included in the model. As your patient groups differed on factors that have shown robust associations with cognition in previous work (i.e., age, DD, EDSS) these should be controlled in your models. If this was not done, the analyses will need to be revisited. If you have controlled for these clinical and demographic variables, you need to make this more explicit.

The analyzes performed were only Pearson's correlation. We did non perform a regression analysis due to the small sample size and the number of variables. All test scores were corrected for demographic variables. No correction was made for clinical variables (such as EDSS and DD) as the course diagnosis (SP vs RR) of MS already accounts for them. All these data have been specified in the manuscript following the reviewer's suggestions.

Given the number of variables, a more conservative approach should be taken to account for multiple comparisons than alpha .05

We agree with the reviewer on this point. However, we did not choose ​​for a more conservative approach as the limited sample size would have led to underestimating an effect that is present but would only be seen with a less conservative approach.

Lines 214-215: try to be more concise in your description of measures that differed vs. those that did not – you could delete “About mood and fatigue scores” and write, “A statistically significant difference was found between groups on fatigue (p=.001), though groups did not differ on anxiety, depression, and anger (all p values >.05).”

As suggested by the reviewer we have rephrased the sentence about this aspect.

Results:

Please clarify if the patient sample was the same as in your prior research

The patients in this study are not the same ones included in the previous work. The two works are part of a larger protocol and divided into two sub-studies. The first aimed at tracing the neurophysiological correlates through MRI, the second aimed at answering the question about the relationship between the form of the disease and the neuropsychological and emotional correlates of the ER. Since the two studios are part of a larger project, we have tried to maintain exactly the same number and distribution.

How was the estimated age of onset determined? Did participants indicate symptoms prior to formal diagnosis? Please clarify.

The disease duration refers to the time since formal diagnosis. We have not considered whether the patients report previous symptoms as we could not have verified the direct link with the pathology and we could not have been able to ascertain their real occurrence.

Table 3: indicate in the title the type of analysis used and report the coefficients and effect sizes, not just the p values (this should also be done in-text) so we can see the direction and magnitude of the effects

According to the reviewer suggestion we have modified Table 3 and included the direction and the magnitude of the effects.

Discussion

The timeliness of your work is nicely highlighted by describing how we have had to rely on the eyes when wearing masks.

Thank you for your kind appreciation.

See comments above re causal statements about brain-behaviour relationships and reframe as correlates/predictors

According to the reviewer's suggestion we have modified this statement in the correlation in several parts of the manuscript.

Figures and Tables:

Consider reformatting Table 1 to be consistent with the presentation of the other tables (i.e., Groups on the top and variables down the side of the table with p values indicated within the row vs. column)

We have reformatted Table 1, as suggested.

I would suggest bolding significant p values to highlight the significant differences

Thanks for the suggestion. We have modified it using the bold.

Round 2

Reviewer 1 Report

Dears,

After corrections made, the manuscript can be published. I think that acronym "MS" should be explained also in the Abstract after the first use.  I don't have further remarks.

Author Response

Thank you for the reviewer's precious work. 

We have followed his/her suggestion and included "Multiple Sclerosis" also in the abstract. 

Reviewer 2 Report

I think the paper is much improved and that the authors described quite well the study limits, so as to allow readers to frame the work and its conclusions. Further language editing could be helpful. Also, the images are of poor quality.

Author Response

We would like to thank the reviewer for his/her precious work. We have modified the quality of one of the immages in order to improve the readability of the manuscript. 

Reviewer 3 Report

See attached responses to authors' replies.

Author Response

We would like to thank the reviewer for his/her careful work.

We have tried to solve all his/her issues. 
